MINIREVIEW
# Bile Acids, Gut Microbes, and the Neighborhood Food Environment—a Potential Driver of Colorectal Cancer Health Disparities

Patricia G. Wolf,[a,b,c] Doratha A. Byrd,[d] Kate Cares,[e] Hanchu Dai,[f,g] Angela Odoms-Young,[a] H. Rex Gaskins,[c,f,g,h] Jason M. Ridlon,[c,f,g,h] Lisa Tussing-Humphreys[a,b,e,g]

[a]Institute for Health Research and Policy, University of Illinois Chicago, Chicago, Illinois, USA
[b]University of Illinois Cancer Center, University of Illinois Chicago, Chicago, Illinois, USA
[c]Department of Animal Sciences, University of Illinois Urbana-Champaign, Urbana, Illinois, USA
[d]Department of Cancer Epidemiology, Division of Population Science, H. Lee Moffitt Cancer Center & Research Institute, Tampa, Florida, USA
[e]Department of Kinesiology and Nutrition, University of Illinois Chicago, Chicago, Illinois, USA
[f]Carl R. Woese Institute for Genomic Biology, Urbana, Illinois, USA
[g]Division of Nutritional Sciences, University of Illinois Urbana-Champaign, Urbana, Illinois, USA
[h]Cancer Center at Illinois, University of Illinois Urbana-Champaign, Urbana, Illinois, USA

**ABSTRACT** Bile acids (BAs) facilitate nutrient digestion and absorption and act as signaling molecules in a number of metabolic and inflammatory pathways. Expansion of the BA pool and increased exposure to microbial BA metabolites has been associated with increased colorectal cancer (CRC) risk. It is well established that diet influences systemic BA concentrations and microbial BA metabolism. Therefore, consumption of nutrients that reduce colonic exposure to BAs and microbial BA metabolites may be an effective method for reducing CRC risk, particularly in populations disproportionately burdened by CRC. Individuals who identify as Black/African American (AA/B) have the highest CRC incidence and death in the United States and are more likely to live in a food environment with an inequitable access to BA mitigating nutrients. Thus, this review discusses the current evidence supporting diet as a contributor to CRC disparities through BA-mediated mechanisms and relationships between these mechanisms and barriers to maintaining a low-risk diet.

**KEYWORDS** bile acids, cancer disparities, colorectal cancer, gut microbes, microbes, social equity, nutrition equity, nutrition

**B**ile acids (BAs) comprise a pool of cholesterol- and microbiome-derived metabolites involved in several important metabolic processes (e.g., cholesterol homeostasis and lipid digestion) (1, 2). Dysregulation of normal BA synthesis and metabolism may be involved in the etiology of colorectal cancer (CRC), the third overall leading cause of cancer death in the United States (3, 4). In a recently conducted prospective, nested case-control study, circulating BAs were strongly, directly associated with risk for incident CRC (5). Furthermore, secondary BAs, derived via metabolism by gut bacteria of primary BAs that escape enterohepatic circulation, are promoters of oxidative stress, inflammation, and DNA damage (6, 7). Therefore, reducing the abundance of some BAs may be beneficial for CRC prevention.

It is well established that dietary and lifestyle exposures are associated with systemic BA concentrations. There is particularly evidence for the role of dietary fat and fiber in BA regulation (2, 8–11). Dietary fat stimulates hepatocytes to secrete BAs, which assist in lipid solubilization and absorption in the small intestine (2, 11). In contrast, *in vitro* studies suggest that fiber binds BAs, reducing BA reabsorption into the terminal ileum and increasing BA excretion in the stool (8, 9). Despite strong biological plausibility for the role of fat and fiber in BA regulation, additional studies that investigate the role of dietary

Address correspondence to Patricia G. Wolf, gondry2@illinois.edu, or Lisa Tussing-Humphreys, ltussing@uic.edu.

The authors declare no conflict of interest.

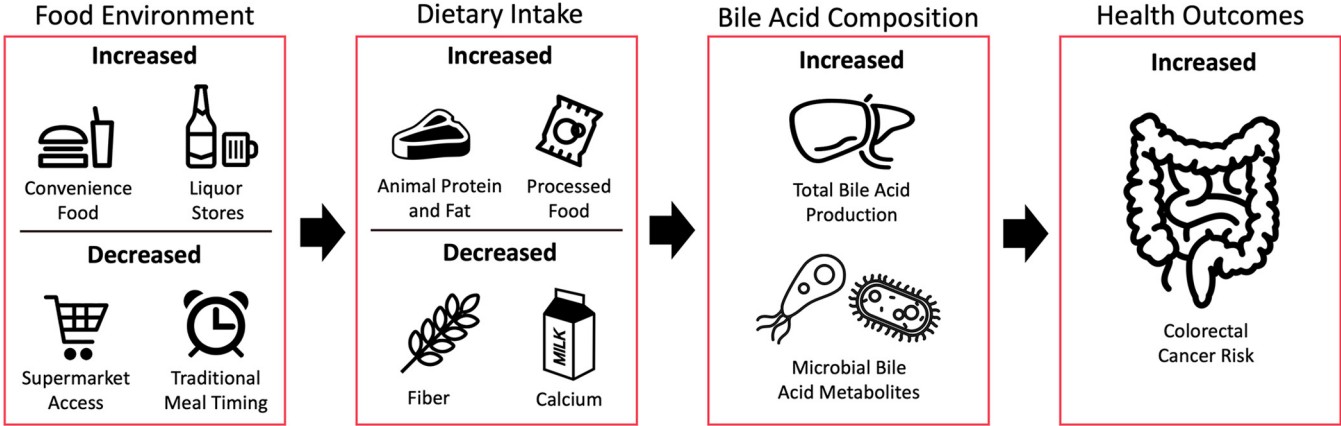

**FIG 1** Food environment and bile acid-mediated colorectal cancer disparities. Groups vulnerable to colorectal cancer disparities are more likely to be exposed to barriers in the food environment to a high-quality diet. This environment may drive dietary behaviors toward nutrients that increase colonic exposure to bile acids and their microbial metabolites, thereby contributing to increased colorectal cancer risk.

components in BA synthesis and metabolism in diverse population-based human studies are critically needed.

The CRC incidence and mortality rate among individuals who identify as Black/African American (AA/B) is approximately 20% and 40%, respectively, higher than that in non-Hispanic white (NHW) Americans (12). Drivers of CRC disparities are complex and likely include a combination of differences in diet, lifestyle, and other socioeconomic, social, cultural, and environmental exposures. Here, we discuss the current evidence supporting diet/lifestyle as a contributor to CRC disparities through BA-mediated mechanisms. We additionally discuss these interrelationships in the context of nutrition equity, which comprises systemic issues serving as barriers to maintaining a low-risk diet (Fig. 1). While these same mechanisms likely contribute to disparities in CRC incidence and mortality observed in other racial/ethnic groups, this review will focus on AA/Bs who bear the highest CRC burden in the United States (3, 4).

## FECAL BILE ACID COMPOSITION AND COLORECTAL CANCER RISK

In human adults, over 20 secondary bile acids are produced as a consequence of microbial biotransformation of primary bile acids. These metabolic modifications alter not only their affinity to nuclear and G-protein coupled receptors but also the hydrophobicity of BAs. The detergent properties of hydrophobic BAs produce membrane perturbations that result in the production of reactive oxygen and nitrogen species (ROS/RNS). This triggers inflammation and apoptotic resistance through endoplasmic reticulum stress, mitochondrial and DNA damage, genomic instability, and transcription of NF-$\kappa$B (13). Therefore, it is unsurprising that BA pool expansion and increased exposure to hydrophobic BAs like chenodeoxycholic acid (CDCA), deoxycholic acid (DCA), and lithocholic acid (LCA) have been associated with CRC development (14–18). However, the proinflammatory impact of hydrophobic secondary BAs may be mitigated by microbial oxidation and epimerization of BA hydroxy groups, producing secondary BA isoforms that may exert antimicrobial and anti-inflammatory effects (19).

Fecal bile acids are predominated by the hydrophobic secondary BAs LCA and DCA (20). Production of these metabolic by-products as a consequence of bacterial 7$\alpha$-dehydroxylation appears to be a key mechanism of colorectal carcinogenesis (7). A recent meta-analysis of studies observed the gut microbial BA inducible (bai) 7$\alpha$/7$\beta$ dehydroxylation operon to be significantly more abundant in metagenomes from colorectal cancer subjects across all cohorts (21). Additionally, in an *N*-methyl-*N'*-nitro-*N*-nitrosoguanidine (MNNG)-induced model of colon carcinogenesis, rectal infusion of LCA and DCA resulted in increased development of colorectal adenomas (22). *In vitro*, LCA and DCA treatment at physiological concentrations produces DNA damage in the colonic cell line HT29 (23), and lysates of cells treated with LCA induce single-strand DNA breaks in intact L1210 cells (24). In addition, DCA induces the extracellular signal-regulated kinase pathway, thereby promoting proteasome-mediated

degradation of the tumor suppressor p53 (25). Importantly, investigators have shown that serum DCA concentrations were higher in men with colorectal adenomas (17) and that fasting DCA concentrations were positively correlated with proliferation in colonic mucosal biopsy specimens (18). Together, these observations implicate the accumulation of hydrophobic DCA and LCA by gut microbiota in the formation of CRC.

## BACTERIAL BILE ACID METABOLISM AND COLORECTAL CANCER DISPARITIES

Dietary intake has been strongly associated with BA synthesis and excretion. For example, Reddy and Wynder compared fecal BA content from North Americans who regularly consumed a plant-based diet or a western-style dietary pattern high in animal protein and fat and low in fruits, vegetables, and whole grains. Compared to a plant-based diet, subjects who consumed a western-type diet had significantly higher fecal concentrations of secondary BA (26). In a follow-up crossover controlled feeding study, consuming a 4-week western diet increased fecal abundance of secondary BA and anaerobic bacteria compared to a low-meat diet (27). This indicated that a western dietary pattern supports the growth of microbes that produce proinflammatory secondary BA as a consequence of their metabolism. In accordance with this, a crossover feeding study investigating short-term dietary changes on the gut microbiome revealed that an animal-based diet (composed of meat, eggs, and cheese) increased fecal concentrations of total and secondary BA as well as short-chain fatty acids (SCFAs) produced from amino acid fermentation. On the other hand, a vegetable-based diet (composed of grains, fruits, legumes, and vegetables) increased fecal concentrations of the protective SCFAs generated via carbohydrate fermentation. In addition, 16S rRNA sequencing revealed the bile-tolerant bacteria *Bilophila wadsworthia* and *Alistipes putredinis* to be among the most enriched taxa representing the fecal microbiome during the animal-based diet. Intriguingly, *B. wadsworthia* produces genotoxic $H_2S$ via metabolism of the sulfur amino acid taurine, which is liberated via bacterial BA deconjugation. Consequently, transcriptome sequencing analysis revealed increased expression of bacterial genes related to BA and sulfur amino acid metabolism on the animal-based diet, including bile salt hydrolase and dissimilatory sulfite reductase (28). Together, these data demonstrate that a western-type dietary pattern, through its effects on BA metabolism, supports a colonic milieu conducive to the formation of CRC.

The observations described above are in accordance with a new line of inquiry that implicates the convergence of diet and microbial BA metabolism as an environmental insult contributing to CRC health disparities. In the United States, AA/Bs have the highest rates of CRC incidence and mortality among the racial/ethnic groups (3, 4). Native South African Blacks have a nearly negligible CRC risk, although rates have been increasing over time in response to dietary changes and increased access to health care screening (29). Intriguingly, a traditional native South African diet is high in fiber and low in animal protein and fat, while AA/Bs are likely to consume a western-type diet. To investigate whether differences in CRC risk are related to diet, a study exploring differences in fecal metabolites and microbial community structure upon diet exchange was conducted. While consuming their typical diet, AA/Bs had significantly higher total fecal BAs and significantly lower SCFAs than native South African subjects (64.4 versus 5.5 $\mu$mol/g, respectively). Dietary exchange produced reciprocal changes in SCFAs and total and secondary BA as well as an abundance of microbial BA metabolizing genes, which corresponded to respective changes in proliferative and inflammatory markers in the colonic mucosa (30). Overall, these findings reinforce that a western-type dietary pattern decreases SCFA production, increases fecal BA abundance, and supports bacteria that induce colonic inflammation through the production of proinflammatory secondary BA metabolites. Furthermore, the data indicate that a dietary intervention is an effective method for changing fecal BA and microbial composition in an effort to lower the inequitable burden of CRC risk among AA/Bs.

An intriguing outcome of this intervention was that dietary exchange produced reciprocal changes in genes related to $H_2S$ production via sulfur amino acid metabolism (30). Increased taurine metabolism may result from greater output of total BA (31) and may also be a consequence of increased tauroconjugation of BA. Consumption of taurine can shift the taurine-glycine conjugation ratio from 1:3 to 10:1, and tauroconjugation of BA serves as

a mechanism of host cysteine balance (7). Therefore, a dietary pattern high in animal protein and fat may produce a procarcinogenic microbial environment through multiple mechanisms, including (i) increased production of total BA and hydrophobic BA induced by fat consumption, (ii) increased tauroconjugated BAs due to protein consumption and subsequent production of $H_2S$ and secondary BAs, and (iii) increased luminal concentration of dietary sulfur amino acids that escape absorption, leading to bacterial $H_2S$ production.

With this in mind, a more recent study examined associations between bacteria that produce $H_2S$ as a factor to explain AA/B CRC disparities in the United States. Investigators examined differences between dietary intake, gut microbial composition, and CRC incidence between AA/B and NHW cancer patients and controls in urban Chicago. This study observed that $H_2S$-producing bacteria were more abundant in AA/B subjects regardless of disease status. Intriguingly, the bile-tolerant bacterium *B. wadsworthia*, which produces $H_2S$ via taurine metabolism, was a significant marker of AA/B CRC. AA/B subjects consumed significantly more protein and fat per 1,000 kcal than NHW subjects, and 16S rRNA gene sequencing of colonic mucosa revealed several $H_2S$ producing genera to be significantly associated with AA/B CRC (32). Together, these findings provide compelling evidence that risk for CRC, and potentially CRC racial inequities, are linked to differences in BA composition, luminal amino acid concentrations, and bacterial metabolite production driven by a western-type dietary pattern.

## DIETARY INFLUENCES ON BILE ACID COMPOSITION AND MICROBIAL BILE ACID METABOLISM

While the described work implies that BA-mediated CRC risk is consistent with certain dietary patterns (vegetarian versus western diet) or major differences in macronutrient composition (high fat versus low fat), it has become clear that these mechanisms may be far more complex. For example, CRC risk is lowest in regions of Africa and south-central Asia (33). However, communities within these regions, like those of Inner Mongolia, have dietary patterns that mirror a western-type pattern (34). It has been posited that this inconsistency is due to underreporting of CRC incidence as a consequence of poor CRC screening access. Indeed, rates of CRC incidence for people living in Inner Mongolia and South Africa have increased with urbanization and education level (29, 35). In addition, urbanization in these and other nonindustrialized societies has shifted dietary patterns to include more market-ready foods higher in saturated fat and simple carbohydrates. These alterations do not change the crude macronutrient content of the overall diet, but the more calorically dense, nutrient-deficient makeup of these items has increased adiposity within the population (36, 37). While cancer risk has not been directly linked with these modifications of dietary pattern, obesity is considered a risk factor for colorectal cancer pathogenesis (38).

These new data illustrate the importance of continued research on the specificity of dietary components as well as how processing and preparation of foods can increase CRC risk. As mentioned, increased intake of sulfur amino acids shifts the BA pool toward tauroconjugation, potentially supporting bacteria that contribute to colonic inflammation through the production of $H_2S$ (7). In addition, while populations that consume a low-fat vegetarian diet have lower CRC risk, a high-fat Mediterranean diet appears to be protective (39). Therefore, the type of fatty acids in the diet and the consumption of BA binding nutrients may be relevant to BA composition and cancer progression. Furthermore, disruptions to host and microbial circadian rhythms due to dietary composition and dietary timing may alter BA production. Thus, the following section will touch upon dietary influences on BA composition and microbial BA metabolism to better understand their combined impacts on the development of CRC (Table 1).

**Fatty acids.** In a systematic analysis of associations between *a priori*-selected dietary components and BA concentrations in the Alpha-Tocopherol, Beta-Carotene Cancer Prevention Study (ATBC), investigators observed that consumption of polyunsaturated fatty acids (PUFAs) and trans fats was associated with increased circulating concentrations of conjugated primary BAs (40). Consistent with this, in a series of studies that investigated the impact of diverse sources of dietary fat on chemically induced carcinogenesis, feeding of diets high in animal

**TABLE 1** Dietary influences on bile acid composition and microbial bile acid metabolism contributions to colorectal cancer development

| | Diet Contribution | Bile Acid Composition | Microbial Metabolites | Host Outcomes |
|---|---|---|---|---|
| **Deleterious** | **Animal Protein**<br>Taurine<br>Cysteine | ↑ Taurine conjugated bile acids | ↑ Secondary bile acids<br>↑ $H_2S$ | ↑ Oxidative and nitrosative stress<br><br>↑ Cell proliferation |
| | **Dietary Fat**<br>Saturated fat<br>n-6 PUFAs | ↑ Total bile acids<br>↑ Phospholipid rich bile | ↑ Secondary bile acids<br>↑ Diacylglycerol<br>↑ Arachidonic acid | ↑ Inflammation<br><br>↑ Apoptotic resistance |
| | **Alcohol** | ↑ Total bile acids | ↑ Secondary bile acids | ↑ Endoplasmic reticulum stress<br><br>↑ Mitochondrial damage |
| | **Meal Timing** | Impaired bile acid synthesis | | ↑ DNA damage<br><br>↑ Genomic instability |
| **Protective** | **Dietary Fiber**<br>Insoluble fiber<br>Soluble fiber | ↓ Total bile acids (Insoluble fiber)<br>↑ Total bile acids (Soluble fiber) | ↓ Secondary bile acids (Insoluble Fiber)<br>↑ Secondary bile acids (Soluble Fiber) | ↑ Fecal bulk<br><br>↑ Luminal viscosity |
| | **Calcium** | ↓ Primary bile acids | ↓ LCA/DCA ratio | ↑ Bile acid precipitation |

fat or n-6 PUFAs (corn oil or safflower oil) consistently increased colonic tumor burden compared to feeding coconut oil or low-fat diets (41). Intriguingly, the fecal BA pool significantly shifted toward the accumulation of DCA, LCA, and 12-keto LCA in rats fed animal fat or n-6 PUFAs (42). This indicates that fatty acids have differential effects on host BA excretion and microbial BA metabolism, leading to the accumulation of hydrophobic secondary BAs that promote tumor incidence. Indeed, a number of studies have demonstrated variations in fecal microbial composition in animals upon treatment with oils high in monounsaturated fatty acids (MUFAs) (olive and canola), n-3 PUFAs (fish and flax), or n-6 PUFAs (corn, safflower, and sunflower) (43, 44). Due to the descriptive nature of these studies, it is difficult to know whether MUFAs and n-3 PUFAs reduce bacterial BA metabolism, but polyphenolic compounds from olive oil have been shown to have antimicrobial effects on bacteria with known bile salt hydrolase activity (45).

In addition, PUFAs increase secretion of phospholipid-rich bile, which may be metabolized by both host and resident bacteria to produce diacylglycerol and arachidonic acid (46–48). Previous work has demonstrated that incubation with biliary phospholipids and DCA stimulated diacylglycerol production by fecal bacteria (47). Diacylglycerol and prostaglandins formed via COX-2-mediated arachidonic acid production stimulate protein kinase C alpha (PKC), a cell signaling enzyme and modulator of cell proliferation (49). Unsaturated $C_{12}$-$C_{20}$ fatty acids can allosterically inhibit COX-2 activity or competitively inhibit the catalytic subunit. In particular, the $C_{20}$ n-3 eicosapentaenoic acid (EPA), abundant in marine oils, reduces COX-2 activity by 50% (50). Accordingly, in a chemically induced model of tumorigenesis, mucosal diacylglycerol kinase and PKC activity were elevated in rats fed a high-fat corn oil diet, while this effect was repressed in rats fed a high-fat fish oil diet (44).

Differential associations observed between fatty acid type and BA composition point to the importance of considering the method of food preparation when studying diet and BA-mediated tumor promotion. A meta-analysis of 42 cohort studies observed reduced risk of CRC among high fish consumers. Notably, this analysis focused solely on fresh fish consumption and excluded intake of fish that was smoked, salted, or fried. Rationale cited by authors included changes to lipid composition from frying oil and generation of carcinogenic compounds through food processing, including nitrates, heterocyclic

amines, and the mutagen 2-chloro-4-methylthiobutanoic acid (51). Consistent with this, several studies have observed that consumption of non-fried fish is associated with reduced pancreatic cancer incidence, while fried fish and shellfish are not (52, 53). Conspicuously, vegetable oils used for deep fat frying are abundant in the n-6 PUFA linoleic acid (54). Thus, consumption of fried foods may change circulating BA composition and support the production of proinflammatory microbial metabolites, including arachidonic acid and secondary BAs that are linked to colorectal carcinogenesis.

**Dietary fiber.** A systematic analysis of 99 studies conducted by the WCRF/AICR concluded there is strong evidence that consuming whole grains and foods high in dietary fiber decreases risk of CRC (55). A more recent systematic review of 221 CRC study meta-analyses confirmed that whole grains were inversely associated with CRC (56). While the mechanisms underlying this association are complex, it has been proposed that these nutrients increase microbial generation of protective short-chain fatty acids and reduce colonic exposure to proinflammatory BAs (30). Increased fecal bulk in response to insoluble dietary fiber may dilute fecal concentrations of secondary BAs and decrease the amount of time that BAs interact directly with the gastrointestinal mucosa due to reduced intestinal transit (57). Furthermore, it has been suggested that partial fermentation of insoluble fibers by gut bacteria reduces the local pH and, thus, inhibits bacterial formation of secondary BAs via $7\alpha$-dehydroxylation (58). Indeed, early work using a chemically induced model of CRC demonstrated that rats fed wheat bran (high in insoluble fiber) had an almost 50% greater fecal output and formed fewer tumors than rats fed a control diet. Correspondingly, concentrations of total and secondary BAs (LCA, DCA, 12-keto LCA, hyoDCA, ursoDCA, and 3-muricholic acid) in rats fed wheat bran were almost half the values observed in control rats, indicating that dilution of BAs reduces their tumor-forming effects (57).

In this same study, rats fed pectin (high in soluble fiber) also demonstrated lower tumor incidence than rats fed a control diet. However, pectin increased fecal output of total BAs and the secondary BAs LCA, DCA, and 12-keto LCA, suggesting alternative mechanisms by which fiber reduces CRC risk (57). Since that time, many studies have demonstrated that direct fiber-BA interactions prevent mucosal exposure by increasing luminal viscosity and fecal excretion of BAs bound within the fiber matrix (59). Increased luminal viscosity is thought to decrease the solubility of BA micelles, reducing their reabsorption in the distal small intestine (59). Consumption of oat bran and citrus pectin increased secretion of total BAs by 42 and 35%, respectively, in subjects with ileostomies (60, 61). A more recent comparison of *in vitro*-digested fibers found that while apple, barley, citrus, lupin, pea, and potato fibers increased digesta viscosity, oat, wheat, and resistant starch fibers did not. However, oat and barley had the highest BA adsorption rates of all the tested fibers, and this adsorptive capacity was associated with BA hydrophobicity. This indicates that the adsorptive effects of dietary fibers are also partially due to molecular interactions with hydrophobic BAs like CDCA and DCA (62). Indeed, studies that have investigated the BA binding capacity of fruits and vegetables have observed that individual BA metabolites have different affinities for dietary fibers (9). These affinities depend partly upon the ratio of insoluble to soluble dietary fibers within the food as well as the particle size, surface area, and molecular structure of the individual fibers (59). Consistent with these mechanisms, the ATBC study observed that dietary fiber intake was inversely correlated with the abundance of circulating BAs (40). However, this observation was inconsistent with two previous studies that observed increased circulating BAs in subjects consuming a high fiber diet (63, 64). It is possible that these discrepancies are due to differences in viscosity, fecal bulk, and BA binding capacity between consumed fibers. Therefore, future research that examines fecal and serum BA concentrations in response to specific dietary fibers is needed to determine which whole foods may have the highest impact on reducing CRC incidence.

**Calcium and alcohol.** Systematic analyses have also concluded that intake of dairy products is inversely associated with CRC, while total alcohol intake is positively associated with CRC (55, 56). Similar to dietary fiber, dairy consumption has been linked to decreased CRC risk by reducing mucosal exposure to proinflammatory BAs. In a chemically induced model of CRC, tumor incidence in response to a high-fat diet was reduced in rats supplemented with calcium (65). In a later study, calcium supplementation precipitated luminal surfactants, increasing total BA excretion and reducing colonic proliferation in rats (66). In

humans, calcium supplementation reduced biliary concentrations of the hydrophobic primary BA CDCA and reduced the LCA-to-DCA secondary BA ratio (67). This was intriguing, as previous studies demonstrated that an LCA/DCA greater than 1 was indicative of CRC (68). Data supporting changes in BA composition in response to dairy consumption have been less convincing; however, this may be due to the dietary fat content of the dairy consumed (69, 70). Indeed, a randomized crossover trial concluded that low-fat dairy consumption had no effect on fecal BAs (70). Thus, follow-up work that examines whether whole-fat dairy feeding modulates BA concentrations is needed to understand the effects of this nutrient on CRC risk.

Heavy alcohol consumption, on the other hand, is consistently associated with increased CRC risk through a number of mechanisms. The metabolites of alcohol metabolism promote genetic damage and instability and induce changes to cellular pathways commonly observed in CRC. Of more recent interest is the effect of chronic alcohol consumption on the composition of circulating BAs and their concomitant associations with cancer development (71). In the ATBC study, alcohol consumption is positively associated with circulating primary and secondary BAs (40). Consistent with this, a recent study investigating chronic ethanol consumption on enterohepatic circulation in rats observed that alcohol abuse (50%, vol/vol, ethanol) increased the abundance of total BAs in all compartments studied (colon, gallbladder, intestine, liver, and plasma). This BA pool expansion was associated with increased expression of genes that upregulate BA production, including *Cyp7a1*, *Cyp27a1*, *Cyp8b1*, and *Baat* (71). Several studies have established changes in gut microbial composition in response to alcohol consumption, most notably an increase in proteobacteria and fusobacteria (72). Coincidentally, bacteria from these phyla, like *B. wadsworthia*, *Escherichia coli*, *Fusobacterium*, and *Desulfobivrio* spp., are among the most commonly associated with CRC (21, 32, 73–75). However, to our knowledge, a study that examines ecological and transcriptional changes in BA-metabolizing bacteria related to alcohol-induced changes in BA composition has not been performed. Given that a recent multiethnic cohort study observed that associations between alcohol consumption and CRC differed by race/ethnicity, this work could contribute to the understanding of mechanisms underlying CRC disparities (76).

**Meal timing.** The circadian cycle is a 24-h biological clock or rhythm that orchestrates physiologic processes in living things, including endocrine, sleep/wake cycles, and energy metabolism (77). It is believed that circadian disruption is conducive to tumorigenesis through cellular metabolic reprogramming, redox imbalance, and chronic inflammation (78). There is also emerging evidence that BAs and the circadian system are intimately linked, given their roles in energy and nutrient metabolism (77). For example, in animals, circadian disruption interferes with BA homeostasis via transcriptional hindrance of the *Cyp7a1* promoter (79). It has been observed that BA concentrations follow diurnal variation with fasting/feeding cycles, and mice deficient in circadian related genes (e.g., *Rev-erb*α) have impaired BA synthesis (80). Moreover, BA-related disorders such as cholestasis, when induced in animals, are associated with increased expression of circadian-related genes, further demonstrating the connection between the circadian system and BA metabolism (81). The gut microbiome is also a key player in this relationship, given its role in BA metabolism and as a circadian organizer in response to host fasting/feeding (82). Demonstrating this connection, in a mouse model, increased gastrointestinal expression of microbial bile salt hydrolase activity was attributed to circadian rhythm dysfunction (83). Bile salt hydrolase cleaves the amino acid side chain of glyco- or tauroconjugated BAs to generate deconjugated BAs (i.e., cholic and chenodeoxycholic acids), which then are subject to further bacterial modification to yield secondary BAs (i.e., DCA and LCA), creating a colonic metabolic milieu conducive to tumor formation (7). Thus, it is possible that a disrupted circadian rhythm alters BA homeostasis and leads to increased production of BA metabolites implicated in colonic inflammation and the promotion of CRC.

## THE NEIGHBORHOOD FOOD ENVIRONMENT, MICROBIAL BILE ACID METABOLISM, AND COLORECTAL CANCER DISPARITIES

As described above, the disproportionate burden of CRC in AA/B individuals may be partially driven by diet-mediated changes in microbial BA metabolism. However, NHW

individuals also consume a western-type dietary pattern, and data regarding dietary differences between groups is inconsistent (84–88). Therefore, it is possible that these disparities are a complex reflection of genetic, environmental, and cultural influences combined with social and structural barriers in the neighborhood food environment that promote the accumulation of proinflammatory BAs. There is overwhelming evidence that minority racial/ethnic groups are more likely to live in areas disproportionately plagued by economic and structural inequalities (89–91). Consequently, low socioeconomic status (SES) is a risk factor for certain cancers, and persons living under the federal poverty line are at highest risk for early death (92). Indeed, a 6-decade review of U.S. cancer statistics demonstrated widening disparities in cancer mortality for subjects who lived in areas of lower SES (93), and a recent study revealed that CRC risk increased with population density, which was inversely correlated with neighborhood SES (94). While these disparities are likely the result of complex exposures that interact to influence CRC risk (e.g., stress, racism, and environmental pollutants), the next section will focus on potential associations between barriers to the neighborhood food environment, microbial BA metabolism, and CRC disparities.

There is considerable evidence that U.S. areas of low SES have a lower density of full-service supermarkets (95–98) and that communities predominately comprised of AA/B individuals were 1.1 miles further, on average, from the nearest full-service supermarket than communities comprised primarily of NHWs (95). Lower SES neighborhoods are commonly predominated by smaller food outlets like convenience stores and bodegas, which have fewer foods in general and fewer healthier options (99, 100). Indeed, a study in 8,462 food stores across 46 states observed that very low-income AA/B and Hispanic communities had a lower ratio of healthful options (e.g., wheat versus white bread) in food stores compared with very high-income NHW communities (ratio of 0.60 and 0.74, respectively) (101). A systematic review of 54 studies revealed that individuals with access to supermarkets or healthy grocers consume more whole foods and have higher diet quality than those without access (102). Indeed, Americans consumed between 11 and 32% more fruits and vegetables with the presence of one additional supermarket per census tract (97), and low-fat dairy consumption correlated directly with the proportion of product available within the zip code (103). Coincidently, a previous investigation in CRC patients and controls demonstrated total servings of dairy negatively correlated with abundance of bacteria that produce genotoxic $H_2S$ (32). Given evidence that both dietary fiber and calcium reduce BA bioavailability to gut microbes, this demonstrates the importance of the proximity of whole foods to improve overall diet quality and mitigate CRC disparities.

In addition to fewer healthful options in local food stores, it has been suggested that stores in lower-SES areas face unique challenges to food quality that may discourage local consumers from the purchase of whole foods high in fiber and calcium (91, 104). Indeed, a comparison of fresh fruit and vegetable access in Detroit neighborhoods observed that mean vegetable quality was significantly lower in predominantly lower SES AA/B neighborhoods (105). Perhaps a more significant health deterrent is the higher rates of foodborne illness experienced by minority and ethnic groups (106, 107). Milk and produce in lower SES neighborhood markets have higher counts of microbes than those from high SES food stores, and those who had previous illness from listeriosis were more likely to have purchased the food from smaller convenience stores than large supermarkets (104). Thus, it is possible that the concerns regarding whole-food quality and safety influence purchasing habits away from nutrients that may mitigate the effects of proinflammatory BAs.

Consumption of convenience foods constitutes the majority of daily caloric intake in the United States. Convenience foods include fast food and foods that are ultraprocessed, a term defining foods that are formulations of industrialized ingredients (108). While recent trends show a decline in calories purchased from ultraprocessed foods in the United States, this reduction is attenuated in low-SES and AA/B populations (109). Ultraprocessed foods are generally calorically dense and deficient in nutrients like calcium and dietary fiber, which may mitigate the accumulation of proinflammatory metabolites produced through microbial BA metabolism (108). Furthermore, the impact of food additives on microbial ecology remains

understudied. For example, the sulfur amino acids taurine and cysteine are commonly added to ultraprocessed foods to act as dietary supplements or dough-conditioning and flavor-enhancing agents (110–112). Increased intake of taurine and cysteine increases tauroconjugation of primary BAs (7, 113). Taurine liberated by bacteria that harbor bile salt hydrolase activity can then be further metabolized by gut bacteria to produce proinflammatory and genotoxic $H_2S$ (7). Therefore, it is possible that increased intake of sulfur amino acids through a diet abundant in ultraprocessed foods increases both direct and indirect $H_2S$ generation, leading to an environment conducive to colorectal carcinogenesis. This puts populations whose food environment is predominantly composed of ultraprocessed foods at particular disadvantage and may serve as a mechanism partially explaining CRC disparities in these groups.

In addition to previously described contributions to food-purchasing behaviors driven by the local grocery environment, lower-SES neighborhoods have a higher density of fast food outlets and liquor stores (99, 114). In a study that examined the neighborhood food environment in four states, higher SES neighborhoods had 3 times fewer places that served alcohol than lower SES neighborhoods (97). This is problematic, as one of the multiple negative impacts of alcohol consumption is that it increases expression of genes that induce bile acid production (71). Additionally, a systematic review investigating differences in accessibility to convenience foods concluded that lower-SES neighborhoods had higher access to fast food outlets and that fast foods were more heavily advertised in AA/B neighborhoods regardless of SES (99). As mentioned above, abundant intake of fast foods may be problematic not only due to the generally high fatty acid content but also to the types of fatty acids being consumed. Fast food restaurants commonly serve foods high in saturated animal fats or that are fried in oils high in unsaturated oleic and linoleic fatty acids (54). These fatty acids increase the total BA pool and shift it toward the production of proinflammatory BAs (41). Hence, the combination of fast food and liquor store density and predatory advertising in low-SES AA/B neighborhoods may drive dietary choices that support a proinflammatory BA profile in populations subjected to CRC disparities.

In addition to the surrounding food environment, individuals working nontraditional work hours may be more susceptible to differences in BA composition associated with CRC. AA/Bs are more likely to arrive at work between midnight and 5 a.m. and between 11 a.m. and noon than other racial/ethnic groups in the United States (115). Individuals working nontraditional hours report irregular sleep and eating patterns as well as increased intake of high-fat and ultraprocessed foods and alcohol, exposures known to affect both the circadian clock and BA metabolism. Work starting late in the evening and early morning is associated with circadian rhythm disruption and is an emerging area in cancer prevention and control research and may also be pertinent to understanding cancer health inequities (78). Thus, mitigating circadian disruption and providing all workers, independent of their work start time, access to healthy foods, areas to prepare healthful meals, and interventions to improve sleep duration and quality may have important effects on circadian rhythmicity, BA metabolism, and, ultimately, colorectal health and cancer prevention.

## CONCLUSIONS

While physiologically important for nutrient digestion and cholesterol homeostasis, the hydrophobic and cell signaling properties of BAs create a proinflammatory environment conducive to colorectal carcinogenesis. Bile acid abundance and composition within the colonic lumen is largely driven by diet-associated secretory responses, microbial metabolism, and direct BA-nutrient interactions (7). Thus, interventions to change dietary choices toward nutrients that reduce the deleterious effects of BAs in the colon may serve as promising solutions to reduce CRC (116). However, prescribing a BA-modulating diet in an effort to reduce CRC disparities does not go far enough, as researchers must take into account the social and environmental context that drives these mechanisms (117, 118). However, previous work examining microbial mechanisms of CRC related to diet lack population heterogeneity, limiting questions regarding structural drivers of disease (119). Dietary choices are often driven by the local food environment, and there is overwhelming evidence that those most susceptible to CRC disparities have inequitable access to high-quality food (120). Thus, collaborative

studies that unite local stakeholders with experts in microbiology, nutrition, and epidemiology in a variety of geographical contexts are needed to investigate links between the local food environment, BA composition, microbial metabolism, and disparities in CRC incidence (121). If associations are observed among these factors, this may serve as a rationale for the expansion of national programs that reduce barriers to BA-mitigating nutrients like the Gus Schumacher Nutrition Incentive Program and the Healthy Food Financing Initiative (120) or localized policies aimed to reduce fast food and convenience store saturation in low SES neighborhoods. Combined, these improvements to the local food environment may alleviate the inequitable burden of CRC incidence and death among AA/Bs in the United States (120).

## ACKNOWLEDGMENTS

This review was funded by T32CA057699 (P.G.W.), R01CA204808 (H.R.G., J.M.R., and L.T.H.), R01CA250390 (L.T.H, J.M.R., and H.R.G.), U54MD012523 (L.T.H., J.M.R., H.R.G., and A.O.Y.), and U54CA202997 (L.T.H.).

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
