## [Reviewer comments · mSystems]

Bile acids, gut microbes, and the neighborhood food environment - a potential driver of colorectal cancer health disparities

Patricia Wolf, Doratha Bryd, Kate Cares, Hanchu Dai, Angela Odoms-Young, H Gaskins, Jason Ridlon, and Lisa Tussing-Humphreys

Corresponding Author(s): Patricia Wolf, University of Illinois at Chicago

Review Timeline:

Submission Date:	September 24, 2021
Editorial Decision:	October 31, 2021
Revision Received:	December 31, 2021
Accepted:	January 4, 2022

Editor: Jotham Suez

Reviewer(s): Disclosure of reviewer identity is with reference to reviewer comments included in decision letter(s). The following individuals involved in review of your submission have agreed to reveal their identity: Nyssa Cullin (Reviewer #3)

Transaction Report:

DOI: <https://doi.org/10.1128/mSystems.01174-21>

October 31, 2021

Dr. Patricia G Wolf
University of Illinois at Chicago
1207 W. Gregory Dr.
Urbana, IL 61801

Re: mSystems01174-21 (Bile acids, gut microbes, and the neighborhood food environment - a potential driver of colorectal cancer health disparities)

Dear Dr. Patricia G Wolf:

Thank you for submitting your manuscript to mSystems. All three reviewers found the manuscript interesting and highlighted its importance. They had several comments and suggestions that should be addressed before acceptance.

Preparing Revision Guidelines

Sincerely,

Jotham Suez

Editor, mSystems

Journals Department
Reviewer comments:

Reviewer #1 (Comments for the Author):

This review by Wolf and colleagues summarizes in a very elegant way the recent literature connecting diet, bile acids (BAs) and the gut microbiome to colorectal cancer (CRC). There is growing interest in mechanistically understanding how bile acids drive cancer pathogenesis, which makes this paper timely. The manuscript looks at these connections from the crucial point of view of health disparities among populations in the US, demonstrating that inequalities in access to quality foods is likely to result in the increase of pro-inflammatory bile acids in the gut, favoring the development of CRC.

I found the paper very well written and insightful and only have a few remarks - see below.

As expected, considering the main theme of the review, BAs and their impact on the host are mostly discussed from a negative angle. I would encourage a broader discussion on the role of bile acids, especially secondary BAs, which includes positive roles on host physiology. For instance, a recent paper (in which I'm not involved) showed that specific secondary BAs are enriched in centenarians, which are characterized by lower levels of chronic inflammation, and that these BAs have anti-microbial effects against pathobionts (10.1038/s41586-021-03832-5).

Section 'Western Dietary Pattern and Colorectal Cancer Risk':

The authors argue that a diet high in animal protein and fat and low in fruits & fibers (defined as the 'Western' diet) associates with higher CRC. While this might be true when looking at industrialized populations, this situation might be more complex. There are many non-industrialized populations worldwide, for instance pastoralists in Africa or Central Asia (e.g. Mongolia), which have diets that fits with this coarse-grained description. Very low rates of chronic diseases are reported in these populations. The counterargument would be that the screening of chronic diseases, such as CRC, might not be as efficient or accurate as in the industrialized world, because of healthcare disparities. In any case, it might be worth discussing the need to include in future research this variety of populations and dietary practices to better disentangle the effects of diets vs. genetics vs. environment on the development of CRC.

I feel that some sections and conclusions are a little too vague and could benefit from more precise descriptions. For instance, in subsection 'Fatty acids', it is said: "This indicates that fatty acids may have differential effects on host BA excretion and microbial BA metabolism, thus leading to the accumulation of proinflammatory secondary BAs". Which FAs? Which BAs? Which ones have been proven to really be 'pro-inflammatory'? Overall, maybe the authors could add more details on which BAs are associated with CRC risks, which ones have been mechanistically connected to CRC in animal models, which ones are connected to insoluble and soluble fiber intake, etc. Maybe referring all these info in a table, or adding more info/data into Table 1?

Still in 'Fatty acids' subsection, it is said "While associations between dietary fat intake and BA-mediated tumor promotion are accepted in models of chemically induced cancer, these trends are inconsistent in humans." Unless I'm wrong, what makes these trends inconsistent in humans is not explained or developed in the paragraph.

In subsection 'Calcium and Alcohol' - "[...] most notably an increase in Proteobacteria and Fusobacteria". There is evidence in the literature linking these two phyla to CRC. Maybe it would be worth mentioning this research, as it would add further support to the argument developed in this section on the impact of alcohol on CRC?

Very interesting review - congrats!

Reviewer #2 (Comments for the Author):

This is a clear and detailed paper outlining the potential role of diet-microbiome interactions in driving CRC risk in African American/Black populations. It does a very nice job of outlining the pathways via which different diet components affect bile acids, which then feedback to affect inflammation and cancer risk. I have relatively few/minor comments. The first is that the section under the heading 'Dietary Influences on Bile Acid Composition and Microbial Bile Acid Metabolism' felt a little repetitive compared to some of the previous material in the paper. I realize this part is much more detailed, but the authors might consider either integrating some of this detail earlier in the paper or removing some of the similar points from earlier in the paper to focus on them more exclusively in this section. I also think it would be helpful to include a little more detail about suggested next steps in terms of research, policy, and community involvement beyond what is in the conclusions. The rest of the paper was so detailed that this part seemed vague in comparison - nothing too long is necessary, but more specifics will strengthen the message.

Reviewer #3 (Comments for the Author):

Wolf et al. present a review entitled "Bile acids, gut microbes, and the neighborhood food environment - a potential driver of colorectal cancer health disparities" as part of a special series highlighting social equity and resolving disparities in microbial exposure.

As a whole, the review is an excellent description of the issues and problems facing specific socio-economic groups and their

increased CRC risk and progression. The authors describe the potential effects of Westernized diets on overall CRC risk and specifically describe BAs and H₂S as modifiable components within the diet. Furthermore, fatty acids, fiber, meal timing, and alcohol consumption are also highlighted as interrelated areas that are highly altered in Black/African American individuals due to overall situational availability.

Overall, Wolf et al. have written a high-quality review including recent mouse studies, human trials, and socio-economic analyses, of a timely subject that should be considered in future scientific studies. One minor suggestion would be to include a brief mechanistic description of how BAs directly cause cancer, perhaps in the introduction. One sentence mentions they "are promoters of oxidative stress, inflammation, and DNA damage." Mentions of the exact mechanisms how these contribute to the hallmarks of cancer, for instance, would solidify the necessity to study such compounds. Two recent reviews cover this well and would provide further information:

(<https://doi.org/10.1152/ajpgi.00261.2020> and <https://doi.org/10.3390/ijms21165786>)

mSystems01174-21

Response to Reviewers: December 5, 2021

Thank you for the opportunity to submit a revision of our article entitled “Bile acids, gut microbes, and the neighborhood food environment - a potential driver of colorectal cancer health disparities,” which we are pleased to resubmit to “*mSystems*”.

We thank the reviewers for their critical and thoughtful review. We have responded to each recommendation below and have updated the manuscript accordingly.

Reviewer #1 (Comments for the Author):

1. As expected, considering the main theme of the review, BAs and their impact on the host are mostly discussed from a negative angle. I would encourage a broader discussion on the role of bile acids, especially secondary BAs, which includes positive roles on host physiology. For instance, a recent paper (in which I'm not involved) showed that specific secondary BAs are enriched in centenarians, which are characterized by lower levels of chronic inflammation, and that these BAs have anti-microbial effects against pathobionts (10.1038/s41586-021-03832-5).

Response: Thank you for providing this reference. Information has been added regarding the function of BAs on host homeostasis and we have provided further clarity regarding the impact of hydrophobic secondary BAs versus secondary BA isoforms enriched in centenarians.

2. Section 'Western Dietary Pattern and Colorectal Cancer Risk':

The authors argue that a diet high in animal protein and fat and low in fruits & fibers (defined as the 'Western' diet) associates with higher CRC. While this might be true when looking at industrialized populations, this situation might be more complex. There are many non-industrialized populations worldwide, for instance pastoralists in Africa or Central Asia (e.g. Mongolia), which have diets that fits with this coarse-grained description. Very low rates of chronic diseases are reported in these populations. The counterargument would be that the screening of chronic diseases, such as CRC, might not be as efficient or accurate as in the industrialized world, because of healthcare disparities. In any case, it might be worth discussing the need to include in future research this variety of populations and dietary practices to better disentangle the effects of diets vs. genetics vs. environment on the development of CRC.

Response: In order to focus the paper on BA modulating nutrients and reduce redundancy, this section has removed. However, we have added a sentence in the conclusions regarding the lack of diversity in populations studied in research to highlight as a need for future work.

I feel that some sections and conclusions are a little too vague and could benefit from more precise descriptions. For instance, in subsection 'Fatty acids', it is said: "This indicates that fatty acids may have differential effects on host BA excretion and microbial BA metabolism, thus leading to the accumulation of proinflammatory secondary BAs". Which FAs? Which BAs?

Which ones have been proven to really be 'pro-inflammatory'? Overall, maybe the authors could add more details on which BAs are associated with CRC risks, which ones have been mechanistically connected to CRC in animal models, which ones are connected to insoluble and soluble fiber intake, etc. Maybe referring all these info in a table, or adding more info/data into Table 1?

Response: Thank you for this suggestion. Finer detail has been added throughout the manuscript in order to clarify associations between CRC, BA modulating nutrients, and specific BAs.

Still in 'Fatty acids' subsection, it is said "While associations between dietary fat intake and BA-mediated tumor promotion are accepted in models of chemically induced cancer, these trends are inconsistent in humans." Unless I'm wrong, what makes these trends inconsistent in humans is not explained or developed in the paragraph.

Response: Agreed. This sentence has been removed.

In subsection 'Calcium and Alcohol' - "[...] most notably an increase in Proteobacteria and Fusobacteria". There is evidence in the literature linking these two phyla to CRC. Maybe it would be worth mentioning this research, as it would add further support to the argument developed in this section on the impact of alcohol on CRC?

Response: Great point! A sentence has been added to highlight the association of bacteria within these phyla and CRC.

Reviewer #2 (Comments for the Author):

1. The first is that the section under the heading 'Dietary Influences on Bile Acid Composition and Microbial Bile Acid Metabolism' felt a little repetitive compared to some of the previous material in the paper. I realize this part is much more detailed, but the authors might consider either integrating some of this detail earlier in the paper or removing some of the similar points from earlier in the paper to focus on them more exclusively in this section.

Response: Thank you for this suggestion. To remove redundancy, we have eliminated the section entitled "Western Dietary Pattern and Colorectal Cancer Risk". This has allowed us to focus the paper on BA modulating nutrients.

2. I also think it would be helpful to include a little more detail about suggested next steps in terms of research, policy, and community involvement beyond what is in the conclusions. The rest of the paper was so detailed that this part seemed vague in comparison - nothing too long is necessary, but more specifics will strengthen the message.

Response: The conclusions have been expanded to include more detail regarding gaps in the literature and next steps.

Reviewer #3 (Comments for the Author):

1. One minor suggestion would be to include a brief mechanistic description of how BAs directly cause cancer, perhaps in the introduction. One sentence mentions they "are promoters of oxidative stress, inflammation, and DNA damage." Mentions of the exact mechanisms how these contribute to the hallmarks of cancer, for instance, would solidify the necessity to study such compounds. Two recent reviews cover this well and would provide further information: (<https://doi.org/10.1152/ajpgi.00261.2020> and <https://doi.org/10.3390/ijms21165786>)

Response: Excellent suggestion! We have replaced the section “Western Dietary Pattern and Colorectal Cancer Risk” with “Fecal Bile Acid Composition and Colorectal Cancer Risk”. This section highlights mechanisms of BA-mediated tumor promotion and provides further clarity regarding what is known about the proinflammatory effects of specific BAs.

January 4, 2022

Dr. Patricia G Wolf
University of Illinois at Chicago
1207 W. Gregory Dr.
Urbana, IL 61801

Re: mSystems01174-21R1 (Bile acids, gut microbes, and the neighborhood food environment - a potential driver of colorectal cancer health disparities)

Dear Dr. Wolf,

Thank you for the opportunity to review your revised manuscript. I enjoyed reading it, and I find that the comments raised by the reviewers have been adequately addressed. Therefore, I am pleased to inform you that your manuscript has been accepted, and I am forwarding it to the ASM Journals Department for publication. For your reference, ASM Journals' address is given below. Before it can be scheduled for publication, your manuscript will be checked by the mSystems senior production editor, Ellie Ghatineh, to make sure that all elements meet the technical requirements for publication. She will contact you if anything needs to be revised before copyediting and production can begin. Otherwise, you will be notified when your proofs are ready to be viewed.

Publication Fees:

We recognize that the video files can become quite large, and so to avoid quality loss ASM suggests sending the video file via <https://www.wetransfer.com/>. When you have a final version of the video and the still ready to share, please send it to mssystemsjournal@msubmit.net.

Sincerely,

Jotham Suez
Editor, mSystems

Journals Department
Phone: 1-202-942-9338